# Exploring the Effects of a Theory-Based Mobile App on Chinese EFL Learners' Vocabulary Learning Achievement and Memory

**Huashan Lu [1], Xingxing Ma [2],* and Fang Huang [3]**

1 School of Foreign Languages, Qingdao Agricultural University, Qingdao 266109, China
2 School of Foreign Languages, Yuzhang Normal University, Nanchang 330103, China
3 School of Education, Shanghai International Studies University, Shanghai 200083, China
* Correspondence: maxingxing@yuznu.edu.cn

**Abstract:** This article intends to investigate the effects of using a mobile app on the vocabulary learning achievement and memory of Chinese tertiary EFL learners. 108 non-English majors took part in the study. Moreover, their perceptions of the app were investigated. By analyzing data from a quasi-experiment, the results show that the experimental group (EG) learned vocabulary at an increased rate of 129.65%, while the control group (CG) did so at an increased rate of 68.4%. In addition, a significant difference was found in vocabulary learning achievement between the two groups, indicating that the app effectively improved students' vocabulary learning. As for remembering the new words, it was discovered that retention of words declined by 11.11% in EG and 27.89% in CG, suggesting that the app was effective in preventing the EG students from rapid memory loss of the words. Among the participants, 72% held a positive attitude toward the app, and 82% expressed that they liked the app in their diaries. Results of this study indicate the usefulness of mobile technology-enhanced L2 vocabulary learning.

**Keywords:** theory-based mobile app; vocabulary learning; vocabulary memory; Chinese tertiary EFL learners; perceptions

## 1. Introduction

Vocabulary learning is crucial for learning a language because language learners cannot understand others or express their opinions without vocabulary. According to McCarthy [1], without the vocabulary to express diverse ideas and meanings, communication in another language cannot be achieved meaningfully even if language learners have a good knowledge of L2 pronunciation and grammar. Indeed, Gass [2] maintains that learning a foreign language is learning its vocabulary.

However, studies show that the English vocabulary size of Chinese university students is at a low level [3–7]. The average vocabulary size of non-English majors is around 2300 words, compared with the requirement of 4500 words to pass the CET 4 exam (College English Test Band 4) in China or the 8000–9000 word families to be capable of various English skills, as proposed by Nation [8] and Schmitt [9]. There might be two reasons to explain the students' low vocabulary level [10]. One reason might involve the disadvantages of word list-based teaching and learning; for instance, this method can make learners bewildered and easily forgetful [11], it is useless for the formation of a semantic network [12], and it can result in the rapid memory loss of vocabulary [13] and low efficiency in memorizing words [14]. Another reason might be insufficient teaching time, which is not adequate to permit English as a foreign language (EFL) learner to achieve a wide vocabulary because vocabulary, learning, and memory are demanding in the EFL setting [15–18].

Mobile-assisted vocabulary learning can help solve the above problems, with special advantages such as accessibility, portability, and personalizability [19]. Many studies [20–24]

have investigated the effects of EFL learners' vocabulary learning through a mobile system. For example, robust research seems to indicate that MALL (mobile-assisted language learning) has positive effects on facilitating vocabulary learning, such as by using flashcards on mobile phones [20], learning English idioms through a Short Message Service on mobile phones [25], and boosting students' engagement in vocabulary learning via smartphones [26]. On the other hand, some less encouraging and even negative results have been revealed in a few studies [27–29]. For example, Stockwell [28] found that learning vocabulary with mobile phones was not better than with desktop computers. Using both techniques, no significant difference was found regarding learners' performances. Additionally, Stockwell [29] noted that learners were unwilling to utilize mobile phones for learning vocabulary. Similarly, Okunbor and Retta [27] revealed that most students who used mobile phones felt the devices were unimportant. Concerning MALL, however, the findings on the effects of mobile appliances vary.

Therefore, further research to explore the effects of MALL on learning vocabulary is needed. More importantly, research concerning the effects of self-developed mobile apps on vocabulary learning achievement is scarce in the EFL context, and there is a necessity to investigate further EFL learners' perceptions of using mobile apps [30–32].

To bridge that research gap, the current study developed a theory-based mobile app, especially for CET 4 vocabulary learning and memory for Chinese university students and tested its effectiveness. In addition, students' perceptions of utilizing the mobile app were examined. To achieve the above purposes, three research questions were generated:

(1) What is the effect of using the mobile app on EFL students' achievement in learning vocabulary?
(2) What is the effect of using the mobile app on EFL students' memory of vocabulary?
(3) What are the EFL learners' perceptions of learning vocabulary with the mobile app?

## 2. Designing a Theory-Based Mobile App for CET 4 Vocabulary Learning

### 2.1. The Theories for Designing the App

Four theories guided the design of the app: ten design principles for MALL [33], dual coding theory [34–36], the cognitive theory of multimedia learning [37,38], and the memory-based strategic framework for vocabulary learning [39]. Table 1 suggested the details of the design principles and adoptions in this study.

Dual coding theory (DCT) suggests an equal treatment of verbal and visual systems and explains the effects of mental images on memory. The verbal system preserves and administers linguistic information, whereas the visual system visual information. Besides, the two systems interact and activate, and this is likely to result in a better recall. Next, vocabulary combined with images is mastered more quickly and memorized more effectively than presented alone [35]. Thus, presenting vocabulary to learners in multiple ways would create a language learning environment, which assisted their learning. Guided by the DCT, target words are illustrated within the app in both audio and visual ways to activate students' two systems to improve their learning and memory.

Based on three hypotheses: dual channels, limited capacity, and active processing, the Cognitive Theory of Multimedia Learning (CTML) concerns how students learn from multimedia presentations. The hypothesis of dual channels implies that humans occupy two information processing channels, which are visual/pictorial and auditory/verbal. The visual/pictorial channel enables learners to get information like pictures or videos with their eyes, and the auditory/verbal channel helps learners to get information like spoken words or sounds through their ears. The hypothesis of limited capacity indicates that the amount of information that learners could process through any channel each time is finite, namely, language learners could only retain a few vocabularies through the auditory/verbal channel in their working memory each time when provided by narration, and that is also true of the active processing channel [38]. To link verbal and visual representations and incorporate them into the learners' minds, instructors should design activities to activate students' long-term memory and lead to working memory. This cognitive process explains

the third hypothesis of active processing. Based on the above, the app includes verbal and visual, printed text, audio words, Chinese definitions, and pictures for students' target word learning. Next, the number of target words to learn is constrained to 10 each time. Additionally, to activate students' cognitive activities from long-term memory to working one, a retrieval section is designed within the app.

**Table 1.** Ten principles and adoptions in this study.

| Ten Design Principles | Adoptions in This Study |
| --- | --- |
| 1. Students' utilizations of mobile apps should be based on learning targets, and pay attention to functions as well as their limitations | Students learn and review target words by themselves after they download the app via smartphones and install it. |
| 2. Constrain the number of tasks and environmental distractions | Word learning tasks: Learning and Retrieval Section (see Section 2.3). |
| 3. Push learners whilst respect their boundaries. | Deadlines are regularly sent to students' smartphones which reminds them to stick to the schedule. |
| 4. Equity should be maintained to the best of one's ability. | Each student of the experiment group holds a mobile device with similar capacities (e.g., like screen size, operating system). |
| 5. Take learners' differences into consideration | All students are from the same province, with similar learning backgrounds and living environments. |
| 6. Know learners' current usage with their mobile devices and their using cultures | Students are Chinese and they share the same cultures of using the smart phones, they mainly use mobile devices for chatting, leisure, and the like. |
| 7. Try best to maintain language learning activities within mobile devices, brief and compendious | The amount of word learning (ten words) each time via the app does not surpass students' working memory capacity. Besides, the target words are presented in an order way within the app. |
| 8. Language learning tasks should be suitable for mobile technology as well as environment, and vice versa. | Short texts in big fonts are suitable for students' reading in mobile devices. |
| 9. Guide and train students to use app effectively for language learning | Students are provided with guidance and training on how to utilize the app for word learning and reviewing |
| 10. Take account of multiple stakeholders | Students autonomously learn and review vocabulary, this way, their apps' usage is not influenced by other people. |

The memory-based strategic framework for vocabulary learning is rooted in findings of memory within psychology and vocabulary acquisition of the second language [39]. The framework consists of the internal memory process and the external strategic behavior. The internal memory process includes four consecutive stages: perceiving the word form, assessing the word meaning, building the word entry, and retrieving the word, which is corresponded to four counterparts of the external strategic behavior: discovering the new word, obtaining the word meaning, mapping the word meaning onto the form, consolidating the word. Based on the framework, in the first stage, visual and audio forms are provided simultaneously to catch students' eyes for the target words in the app. In the second stage, students obtain the meanings of target words with the help of Chinese definitions and example sentences. In the third stage, students are supplied with two multiple-choice exercises: 'choose the correct meaning for a given word' and 'select the correct word for a given meaning', for their mapping meanings of target words onto forms. In the fourth stage, spelling exercises are offered for students to consolidate target words in their memory.

### 2.2. Target Words

The experiment, which took place over 8 weeks during the 2021–2022 academic year, selected 80 target vocabulary words from CET-4 based on frequency. To determine the 80 target words, the researcher compiled a corpus covering CET 4 test papers (2014–2021).



Next, an online word frequency counter was used for counting the word frequency in the corpus. Then, the 80 most frequently used CET 4 words were selected as target words (Supplementary Materials Table S1) and verified regarding three CET 4 vocabulary books [40–42].

### 2.3. Mobile App Design

The mobile app conceived by the researchers consisted of nine steps. Step one: the homepage of the app pops out after students log in via their student ID numbers. Step two: the interface with eight learning sections pops up. Based on the eight-week schedule, students click on the corresponding learning section in ascending order. Step three: in the first week, ten words appear on the interface for each learning section. Step four: the retrieval sections interface will be shown after students finish the learning section and click on the retrieval section corresponding to keep learning on the homepage. Step five: retrieval section containing three columns of exercises appears when the student clicks on it. The first column is designed for students to select the correct meaning for a given word; the second column asks them to select the correct word for a given meaning, and the third column requires students to spell the correct word for a given meaning. Step six: the page for selecting the correct meaning for a given word emerges when students click on the first column. Step seven: the page for selecting the right word for a given meaning is shown after students finish the first column of the exercise and tap on the second column. Step eight: the page for spelling the right word for a given meaning pops up after students finish the second column of the exercises and click on the third column. Step nine: upon completing the third column exercise, students are automatically logged out of the app.

### 2.4. Vocabulary Content Design

The target word is supplied with its phonetic symbols, part of speech, and Chinese meaning, an example sentence, a picture, exercises, and an audio file. The Chinese meaning, phonetic symbols, and part of speech are taken from the Oxford Advanced Learner's English–Chinese Dictionary (2014 version), and the sample sentences and exercises, are taken from three CET 4 vocabulary books [40–42]. The pictures, as well as audio files, are taken from Baidu (the Chinese internet search engine). Lastly, all the above materials relating to the target words are imported into the mobile app, and then the learning and retrieval sections are added. To ensure that the meanings of the target words are clear to all students, the above procedure is used. The nine steps of the app are designed for both the experimental group (see Section 2.3) and the control group (see Supplementary Materials Table S2).

## 3. Research Methodology

### 3.1. Participants

The study population was 850 newcomers at a Chinese university. Two classes with 108 non-English majors were selected as samples based on convenience and availability. The demographic information of the two classes is presented in Table 2.

Table 2 shows that the two classes shared similar characteristics, including the number of students, age, gender distribution, and the length of time spent learning English. The initial English proficiency levels of the two classes were measured with the NCEE (National College Entrance Examination) in China, which is famous for being highly reliable and valid. The result of an independent-sample *t*-test shows that there was no significant difference between the two classes in terms of their initial English proficiency levels ($p = 0.857 > 0.05$), showing that they had similar levels of proficiency before the experiment. Thus, the secretary class was selected as the experimental group (EG) randomly while the Chinese language class was the control group (CG).

**Table 2.** Demographic information of two classes in this study.

| Classes | Number | Mean Age (Years) | Gender | English Learning Experience | English Proficiency (Mean (SD)) | Sig. |
|---|---|---|---|---|---|---|
| Secretary class | 50 | 19 | Male = 22 Female = 28 | 10–11 years | 94.07 (0.45) | 0.857 |
| Chinese language class | 58 | 18 | Male = 25 Female = 33 | 10–12 years | 94.52 (0.47) | |

As presented in Table 3, the mean vocabulary size for the EG was 2006.15 (SD = 597.258), and for the CG is 2110.34 (SD = 578.470). The independent-sample *t*-test shows no significant difference between the two groups ($p = 0.317$). Furthermore, the vocabulary size for each group is considerably less than the 4500 words required for CET-4 in China. Outcomes indicate that the two groups were located at a relatively low level in their vocabulary size, showing that both groups lack sufficient vocabulary and need to learn more to pass CET-4. Accordingly, the students in two groups were considered qualified for this experiment.

**Table 3.** Comparison of students' vocabulary size in EG and CG.

| Group | Number | Mean | SD | t-Value | Sig. (2-Tailed) |
|---|---|---|---|---|---|
| EG | 50 | 2006.15 | 597.258 | 14.258 | 0.317 |
| CG | 58 | 2110.34 | 578.470 | | |

As shown in Table 4, the mean of the pre-test for EG was 147.65 (SD = 26.371), with scores ranging from 89 to 191, and that for CG was 143.82 (SD = 22.707), with scores ranging from 85 to 184. This demonstrates that the target words were new to the students of both groups. No significant difference was identified through an independent samples *t*-test between the mean values for the EG and the CG ($p = 0.584$). This suggests that the two groups were at a similar level in their target word knowledge. Hence, the target words selected were suitable for two groups.

**Table 4.** Comparison of students' vocabulary learning between EG and CG in pre-test.

| Test | Number | Group | Score | | Mean | SD | t-Value | Sig. (2-Tailed) |
|---|---|---|---|---|---|---|---|---|
| Pre-test | 50 | EG | Highest Lowest | 191 89 | 147.65 | 26.371 | 38.229 | 0.584 |
| | 58 | CG | Highest Lowest | 184 85 | 143.82 | 22.707 | | |

Significance level $p < 0.05$. Total score on target words knowledge test is 400 points.

### 3.2. Research Instruments

A vocabulary size test was used as a reliable and valid instrument (Supplementary Materials Section S1) [43] with a high reliability of 0.96 [44] and thus, was chosen to measure vocabulary size in the pre-test. To assess vocabulary learning achievement and memory, the vocabulary knowledge scale (VKS) (Supplementary Materials Section S2) [45] was selected since it has a reliability of 0.89 [46] and is effective in measuring vocabulary learning achievement and memory [47].

A questionnaire concerning students' perceptions of the app was adapted from Chang et al. [48] and Wang [49] (Supplementary Materials Section S3). Its item-objective congruence (IOC) was rated as 0.875 by three experienced English experts, indicating that the questionnaire contents were valid. To test its reliability, it was piloted in a first-year non-English major class and was found to be reliable because Cronbach's alpha is 0.913.

A semi-structured interview (see Supplementary Materials Section S4) based on the questionnaire was used. The validity of the interview questions was confirmed by three

English experts who calculated an IOC index of 0.88. To ensure an appropriate number of interviewees, the Alberta Municipal Health and Safety Association's criteria [50] were followed, resulting in the selection of 19 representative interviewees through purposive sampling. The selection was based on their answers in the questionnaire and diaries.

Students from the two groups kept diaries to record the time they engaged in learning and provide feedback on their vocabulary learning every week. Both groups reviewed each package of 10 words after 9 h, 1 day, 2 days, and 6 days from the first time they learned them. Table 5 shows the format of a diary.

**Table 5.** Format of diary for experimental and control groups.

| **Name:** | | **Student ID:** | | **Major:** | |
|---|---|---|---|---|---|
| 5 times/week | 1st time<br>Suitable time for learning words | 2nd time<br>9 h<br>from first time | 3rd time<br>1 day from first time | 4th time<br>2 days<br>from first time | 5th time<br>6 days<br>from first time |
| 10 words/time | | | | | |
| Start to finish time<br>My feelings about learning and reviewing words in weeks 2–9 | | | | | |

### 3.3. Experimental Procedure

In week 1, the objectives of this study were illustrated to the students in both groups in a classroom setting. All participants voluntarily signed a consent form (EG, Supplementary Materials Section S5; CG, Supplementary Materials Section S6). Following this, the participants of both groups underwent the VST and VKS as pre-tests in the researcher's presence. In the same week, the EG was introduced to the mobile app, which was installed on their smartphones. Meanwhile, a list of the same 8 packages was given to the CG on paper sheets (Supplementary Materials Table S2).

In week 2, students in two groups were instructed to begin learning every package on Monday for the first time, and then to review it outside of class 4 times within a week for 8 consecutive weeks.

In week 10, both groups took the same VKS used for the pre-test but in a different order. In the same week, the EG completed the questionnaire, and 19 participants were interviewed by the researcher.

In week 14, both groups underwent the VKS as a delayed post-test, but in a different sequence. Ebbinghaus' theory (cited in [51]) suggests that details should be retained in long-term memory with minimal memory loss after 4 weeks. Hence, the delayed post-test was administered 4 weeks after the post-test.

### 3.4. Data Analysis

The VKS from the pre-test data was described using mean and standard deviation (SD). An independent-samples *t*-test was employed to determine if there was a significant difference in target word knowledge between the two groups by analyzing the mean of their target word knowledge. The data analysis was performed using SPSS 26.0.

In response to the first research question, two *t*-tests were conducted. The paired-sample *t*-test was utilized to compare the average word knowledge score of each group before and after the experiment. Meanwhile, the independent-sample *t*-test was employed to investigate the difference in the mean score of word knowledge between the two groups after the experiment.

In response to the second research question, the paired-sample *t*-test was applied to compare the mean score of word knowledge for each group between delayed post-test and post-test. The independent-sample *t*-test was used to determine if there was a difference in the mean score of word knowledge between the two groups on the delayed post-test.

In response to the third research question, researchers combine the questionnaire results with findings from the semi-structured interviews and diaries. Firstly, descriptive statistics were used to determine the percentage of each item in the questionnaire data. Secondly, the second author transcribed the semi-structured interviews word by word, including translating any Chinese expressions into English. Other authors helped to evaluate responses line by line to identify categories, which were then developed into final themes through a coding process. The summarized themes provided in-depth information to answer the third research question, along with the questionnaire data. Thirdly, the diary data were numerically coded based on the students' ID numbers and then analyzed using open coding [52] and axial coding [53] techniques, which involved coding and synthesis. The resulting categories and themes were applied to verify the answers to research questions 1 and 2 and to further answer research question 3.

## 4. Results

### 4.1. Effects of Applying the Mobile App on EFL Students' Achievements in Learning Vocabulary

As presented in Table 6, the mean score in the post-test was much higher for the EG (M = 339.08; SD = 41.652) than for the CG (M = 242.20; SD = 34.550). Table 6 also showed a statistically significant difference ($p = 0.001 < 0.05$) between the two groups. This denotes that students learned more words via the app than from the paper-based word list, which indicates that the mobile app effectively improved their achievements in learning vocabulary. This answered the first research question combined with the results below.

**Table 6.** Comparison of students' scores on vocabulary learning between EG and CG in Post-Test.

| | Experimental Group (n = 50) | | Control Group (n = 58) | | MD | t |
|---|---|---|---|---|---|---|
| | M | SD | M | SD | | |
| Score on vocabulary learning in Post-Test | 339.08 | 41.652 | 242.20 | 34.550 | 96.88 | 41.561 * |

* $p < 0.05$.

To determine the potential increase in vocabulary achievements, a paired-sample *t*-test was employed to compare the pre-test and post-test performances of the two groups, respectively, as demonstrated in Tables 7 and 8 below.

**Table 7.** Comparison of scores on vocabulary learning between pre- and post-test in CG.

| | Post-Test (n = 58) | | Pre-Test (n = 58) | | MD | t |
|---|---|---|---|---|---|---|
| | M | SD | M | SD | | |
| Scores on Vocabulary learning in CG | 242.20 | 34.550 | 143.82 | 22.707 | 98.38 | 11.203 * |

* $p < 0.05$.

**Table 8.** Comparison of scores on vocabulary learning between pre- and post-test in EG.

| | Post-Test (n = 50) | | Pre-Test (n = 50) | | MD | t |
|---|---|---|---|---|---|---|
| | M | SD | M | SD | | |
| Scores on Vocabulary learning in EG | 339.08 | 41.654 | 147.65 | 26.371 | 191.43 | 12.258 * |

* $p < 0.05$.

As seen from Table 7, the mean scores on vocabulary learning in CG changed from 143.82 in the pre-test to 242.20 in the post-test, which was an achievement increase by 98.38 (about 68.4%). Using a paired-sample *t*-test, a significant difference in terms of students' mean scores on vocabulary learning between the pre-test and post-test was identified ($p = 0.000 < 0.05$). These indicated that achievements of the target words significantly improved in CG.

Meanwhile, Table 8 shows that the mean score of EG ascended from 147.65 in the pre-test to 339.08 in the post-test, a great increase of 191.43 (129.65%). Further, the paired-sample *t*-test indicated a significant difference between students' mean scores on vocabulary learning in the pre-test and post-test ($p = 0.001 < 0.05$). These showed that achievements of target words significantly improved in EG.

### 4.2. Effects of Using the Mobile App on EFL Students' Memory of the Vocabulary

As illustrated in Table 9, the mean score of the delayed post-test was much higher for EG (M = 305.18; SD = 49.153) than for CG (M = 174.65; SD = 36.843). Further analysis through an independent-samples *t*-test found a significant difference in the delayed post-test between the EG and the CG ($p = 0.002 < 0.05$). This signifies that students who used the mobile app could memorize more words than those who did with the paper-based word list, suggesting that the app was effective in helping students retain the words. This answered the second research question with the results below.

**Table 9.** Comparison of scores on vocabulary learning between EG and CG in delayed post-test.

| | Experimental Group (n = 50) | | Control Group (n = 58) | | MD | t |
|---|---|---|---|---|---|---|
| | M | SD | M | SD | | |
| Scores on vocabulary learning in Delayed post-test | 305.18 | 49.153 | 174.65 | 36.843 | 96.88 | 33.536 * |

\* $p < 0.05$.

For a better understanding of how much target words knowledge was retained in memory for both groups, a paired-sample *t*-test was applied to compare the delayed post-test and post-test performances of the two groups, respectively, as seen in Tables 10 and 11 below.

**Table 10.** Comparison of scores on vocabulary learning between delayed post-test and post-test in CG.

| | Post-Test (n = 58) | | Delayed Post-Test (n = 58) | | MD | t |
|---|---|---|---|---|---|---|
| | M | SD | M | SD | | |
| Scores on Vocabulary learning in CG | 242.20 | 34.550 | 174.65 | 36.483 | 67.64 | 11.203 * |

\* $p < 0.05$.

**Table 11.** Comparison of scores on vocabulary learning between delayed post-test and post-test in EG.

| | Post-Test (n = 50) | | Delayed Post-Test (n = 50) | | MD | t |
|---|---|---|---|---|---|---|
| | M | SD | M | SD | | |
| Scores on Vocabulary learning in EG | 339.08 | 41.654 | 305.18 | 49.153 | 33.90 | 11.369 |

As revealed in Table 10, students' mean scores on vocabulary learning in CG descended from 242.20 in the post-test to 174.65 in the delayed post-test, with a decrease by 67.55 (27.89%). Through a paired-sample *t*-test, a significant difference was seen ($p = 0.000 < 0.05$). This meant that the CG students' memory of the target words declined considerably.

On the other hand, as shown in Table 11, the mean scores of students in EG decreased from 339.08 in the post-test to 305.18 in the delayed post-test, which was a reduction by 33.90 (11.11%). By a paired-sample *t*-test, no significant difference was identified concerning the mean score of the delayed post-test and that of post-test ($p = 0.247 > 0.05$), which indicated that the EG students' memory of the target words retained solidly.

### 4.3. EFL Students' Perceptions of Learning Vocabulary via the Mobile App

Based on the students' responses to a questionnaire, outcomes were required employing a descriptive analysis. The results were then divided into five categories: 1, attitude towards using the app; 2, perceived convenience; 3, perceived ease of use 4, perceived usefulness; 5, intention to continue usage, derived from Chang et al. [48] and Davis [54]. Next, Table 12 provides a summary of their responses.

As shown in Table 12, according to their responses to items 4 and 5, 36 students (72%) had a positive attitude about using the app. Among the students, 42 (84%) agreed that it was convenient for them to use the app for vocabulary learning, as seen from their responses to items 2 and 11. In addition, 37 respondents (74%) announced that the app made vocabulary learning easy, as illustrated by their responses to items 1, 3, and 12. Moreover, 31 students (62%) believed the app was useful for learning and retaining vocabulary, as shown by the responses to items 6, 7, 8, 9, 10, 13, 14, and 15. Finally, 40 students (80%) expressed intention to continue using the mobile app to learn vocabulary in the future, as demonstrated by their responses to item 16.

**Table 12.** Questionnaire responses (number and percentage).

| Category (n, Percentage) | Statements * | Strongly Disagree and Disagree | Undecided | Agree and Strongly Agree |
|---|---|---|---|---|
| Attitude toward using the app (36, 72%) | 4. It is a good method to learn vocabulary via the app. | 4, 8% | 8, 16% | 38, 76% |
| | 5. I prefer the app to the traditional wordlist for learning vocabulary. | 6, 12% | 10, 20% | 34, 68% |
| Perceived convenience (42, 84%) | 2. Learning vocabulary using the app is convenient since I can select time and places to learn words. | 2, 4% | 10, 20% | 38, 76% |
| | 11. I think the app makes vocabulary learning much more convenient outside the classroom. | 0, 0% | 4, 8% | 46, 92% |
| Perceived ease of use (37, 74%) | 1. I think the vocabulary learning app is easy to use. | 3, 6% | 7, 14% | 40, 80% |
| | 3. The app makes vocabulary learning easier for me, compared with a wordlist learning. | 5, 10% | 18, 36% | 27, 54% |
| | 12. I feel learning the words easier based on the images and example sentences in the app. | 3, 6% | 4, 8% | 43, 86% |
| Perceived usefulness (31, 62%) | 6. The vocabulary learning app motivates me to learn new words. | 6, 12% | 16, 32% | 28, 38% |
| | 7. I think the app is useful to learn vocabulary. | 2, 4% | 9, 18% | 39, 78% |
| | 8. The Learning sections in the app help me learn vocabulary more effectively. | 0, 0% | 9, 18% | 41, 82% |
| | 9. The immediate feedback in the app can push me to monitor and adjust my words learning. | 3, 6% | 12, 24% | 35, 70% |
| | 10. The Retrieval sections in the app enable me to review and remember the vocabulary well. | 0, 0% | 18, 36% | 32, 64% |

**Table 12.** *Cont.*

| Category (n, Percentage) | Statements * | Strongly Disagree and Disagree | Undecided | Agree and Strongly Agree |
|---|---|---|---|---|
| | 13. I think the sample sentences in the app can consolidate words' knowledge. | 5, 10% | 16, 32% | 29, 58% |
| | 14. I think the vocabulary learned via the app is not easily forgotten. | 7, 14% | 29, 58% | 14, 28% |
| | 15. The sample sentences help me learn how to use the words appropriately. | 3, 6% | 18, 36% | 29, 58% |
| Intention to continue usage. (40, 80%) | 16. In the future, I will continue to use the app to learn vocabulary. | 2, 4% | 8, 16% | 40, 80% |

\* Statements were adapted from Ma [55].

For the semi-structured interview, by thematic analysis, three themes (fondness for, advantages of, and challenges of the app) emerged from five categories, which were grounded on the interviewees' responses. The five categories are presented in Table 13.

**Table 13.** Categories summarizing interviewees' responses.

| Themes | Categories (Number, Percentage) | Interviewees' Responses (S for Student) |
|---|---|---|
| Students' Fondness | 1 Fondness for the app (18, 94.7%) | S17: . . . I like the app for it has advantages: convenience, usefulness in helping my vocabulary learning and memory . . . |
| Perceived advantages | 2 Usefulness/Helpfulness (19, 100%) | |
| | - in understanding and expanding words | S11: . . . the pictures beside target words make me better understand them . . . |
| | - in remembering words | S14: . . . the example sentences make me memorize words easily . . . |
| | - in improving listening | S6: . . . the audio files of the app can improve my listening . . . |
| | - in adjusting word learning | S15: . . . the app can adjust my traditional way of learning words . . . |
| | - in correcting pronunciation | S18: . . . the audio files guide me to read vocabulary right and correct my mispronunciation of words in the app . . . |
| | - in recalling words | S16: . . . when I had difficulty in doing the spelling exercises, the images and example sentences would help me recall . . . |
| | - in practicing four skills | S8: . . . the app can facilitate my English comprehensive skills including listening, speaking, reading, translation . . . |
| | - in arousing interest in learning words | S2: . . . the way of presenting words and exercises could trigger my interest in vocabulary learning through the app . . . |
| | 3 Convenience (19, 100%) | |
| | - time | S6: . . . I can make use of the app to learn vocabulary whenever I want . . . |
| | - place | S4: . . . no matter where I am, I can learn target words . . . |
| | - weight | S3: . . . Holding a smartphone all the time is not a burden for me . . . |
| | 4 Innovation (17, 89.4%) | |
| | - in the way of learning words | S5: . . . it is innovative to learn vocabulary using an app . . . |

**Table 13.** *Cont.*

| Themes | Categories (Number, Percentage) | Interviewees' Responses (S for Student) |
|---|---|---|
| | 5 Drawbacks (4, 21.1%) | |
| Challenges | - irrelevant information popping out | S7: . . . advertisements show up now and then when I am learning words via the app . . . |
| | - design of exercises | S9: . . . the exercises should be designed richer so that vocabulary can be learned all-round . . . |
| | - network connection | S1: . . . WiFi is still tested on campus, so it is unstable to connect network . . . |
| | - covering the words in textbooks | S12: . . . shortage of the current app is not covering my English textbook, or I can learn more vocabulary with the app . . . |

The interview data show that 18 out of 19 interviewees were satisfied with the app, mainly because of its advantages, which include the second to fourth categories. For example, interviewee No.17 said, " . . . I like the app for it has advantages: convenience, usefulness in helping my vocabulary learning and memory . . . " This supports the students' responses to the questionnaire concerning attitudes. 100% of the interviewees believed that the app was useful or helpful concerning 8 aspects and considered it convenient for learning and reviewing words in three aspects. Furthermore, 17 interviewees mentioned the app as an innovative way to learn vocabulary. For instance, interviewee No.2 said, " . . . it is innovative to learn vocabulary using an app . . . ". However, four interviewees noted the app's drawbacks, for instance, irrelevant information popping up, the design of the exercises, the network connection, and the coverage of the words in the textbooks.

A total of 400 diary entries were collected from students of the EG after the 8-week experiment, which were encoded as follows: diaries (D) per week (W) in EG were encoded by a student's ID numbers, like D3W5EG, D4W6EG, D2W4EG, etc. For example, code: D3W5EG means the third student's diary written during the fifth week from EG. The diaries written in Chinese were translated into English by the researchers first and then cross-checked by two experienced English teachers. Next, the English translations were categorized into themes. Table S3 (see Supplementary Materials Section S7) shows the results concerning the length of time, the time spent studying, and the feelings of students from EG.

The data from the diaries (Supplementary Materials Section S7) show that the length of time students spent on the 10 words was 15 min per session making a total of 75 min per week. Their study periods varied from early morning to late at night. Briefly, the most popular study times for using the app (from highest to lowest) were late at night (45, 90%), during the going-out period (40, 80%), in the early morning (38, 76%), during meals (31, 62%), and between classes (30, 60%).

Ten categories regarding the merits of the app were formed, as follows:

(1) 82% of students preferred the app, indicating that they believed it was convenient, fun, effective, easy, reasonable, satisfactory, and useful in learning and memorizing vocabulary via the app. For example, student D32W9EG wrote: "After using the app to learn vocabulary, I can retain them solidly without tiredness, so I like it". These findings reconfirm the students' positive attitudes toward the app based on the questionnaire and interviews.

(2) For 90% of students, it was convenient for learning and reviewing words. For instance, student D37W4EG wrote: "Whenever and wherever I want to learn or review vocabulary, I can use the app".

(3) For 82% of students, the app effectively improved their vocabulary memory. For example, student D16W4EG wrote: "The learning section and retrieval section of the app could consolidate my memory of the words".

(4) For 88% of students, it was reasonable to learn 10 words through the app independently following the schedule five times per week.

(5) 74% of students were satisfied with the app, saying they could feel their vocabulary increase after using it. For example, student D4W5EG wrote: "I know more words than before week after week, so I am very content with the app".

(6) 76% of students referred to learning and retaining words as fun. For instance, student D8W7EG wrote: "To learn and remember target words through the app bring me much fun".

(7) 68% of students said that it was easy to keep the vocabulary in mind by using the app. For example, student D25W8EG wrote: "it is not difficult for me to retain them after using the app to learn target words".

(8) 84% of students considered the app beneficial in correcting the mispronunciation of words, expanding their interest in learning English, retaining their vocabulary, and enhancing their confidence in using English. For instance, student D39W6EG wrote: "it is useful for me to learn and remember vocabulary fast with the app".

(9) 80% of students showed a willingness to continue using the app. For example, student D34W3EG wrote: "If possible, I will continue to utilize the app for learning vocabulary in the future".

(10) 20% of students mentioned that they had difficulty using the app while adhering to a schedule during their military training concerning remembering the exact pronunciations and meaning of every word and recalling them accurately.

## 5. Discussion

### 5.1. Effects of the App on Vocabulary Learning Achievement

The results of this study show that the students' achievements in vocabulary learning via the app were effective. This corresponds to Chen et al.'s study [56], which reported that EFL students using a mobile app obtained higher scores than those using traditional methods for learning vocabulary. Four reasons may account for the significant achievements of students in this experiment.

First, the mobile app's multimedia environment is advantageous in helping students to learn vocabulary. Combining pictures and text makes it easier for learners to comprehend words. It boosts their learning motivation because it allows them to choose and connect information from visual and verbal sources and thus helps them construct [57]. According to Matsuoka and Hirsh [58], gaining a deep understanding of vocabulary within a multimedia environment can enhance the transfer of knowledge to real-world contexts, and have a lasting impact on language learners. They also argue that using a multimedia environment helps L2 learners to remember words more efficiently and is more effective in improving recall than presenting words in isolation. These findings are consistent with Rusanganwa's [59] study, which found that presenting words through a multimedia approach incorporating text, sound, and pictures can accelerate vocabulary acquisition and improve retention.

Second, immediate corrective feedback provided by the app is advantageous in helping students adjust their vocabulary learning techniques and retain the correct meanings and word forms. Soria et al. [60] reported that timely corrective feedback helps to develop students' vocabulary comprehension, monitor their learning progress, rectify incorrect guesses, and ultimately the correct information was stored in their long-term memory. Likewise, Mollakhan et al. [61] pointed out that timely corrective feedback was effective in helping Iranian English learners identify and correct errors when learning new words. Similar findings were suggested by Roediger and Butler [62].

Third, the app's ability to foster significant improvements in vocabulary learning might be attributed to students' enjoyment. Based on Green [63], there is a positive correlation between the enjoyment of an activity and its effectiveness in learning. According to Sandberg et al. [64], students suggested that using their app was enjoyable and entertaining. This further motivated them to use it during their free time and improved their

English language learning. Hsu et al. [65] indicated that EFL students who enjoyed using mobile apps experienced increased levels of relaxation and paid more attention during vocabulary learning.

Finally, audio materials in the app help students achieve correct pronunciation, which also benefits vocabulary learning. Min [66] suggested that many EFL learners encountered difficulties in vocabulary acquisition due to incorrect pronunciation which leads to misspellings. Kaplan–Rakowski and Loranc–Paszylk [67] suggested that improving pronunciation through audio recordings enhances the retrieval and delivery of words for EFL learners, given that they can learn the words more easily by perceiving the acoustic and orthographic similarities. Similarly, Karousou and Nerantzaki [68] demonstrated that students who listened to audio materials and learned phonology were more adept at recalling and retrieving the lexical items.

### 5.2. Effects of the App on Remembering Vocabulary

This study suggests the app successfully sustained students' word recall, which echoes that of Kohnke et al.'s study [69] revealed that a mobile app effectively boosted students' vocabulary retention. Possible reasons are presented below.

First, spacing out the vocabulary review in the app's retrieval sections may enhance students' word memory. Namaziandost et al. [70] stated that spaced repetition of vocabulary positively impacted students' vocabulary knowledge transfer, from short-term to long-term memory. Performing exercises in the app while spacing out the review is also beneficial as they contribute to effective vocabulary retention [29]. Ma [39] discovered that each time students practiced retrieving words for review, it strengthened their memory traces for the words. Retrieval exercises for vocabulary review tend to yield greater gains in long-term memory than repeated learning [71].

In addition, the dual coding system for words results in improved recall. Lin [72] found associations of verbal and visual modes were highly effective in recalling relevant information, as cues can still be accessed when the other is lost. Boers et al. [73] stated that presenting words using two or more modes is more attention-grabbing for EFL students, leading to better retention. They further explained that using both visual and verbal illustrations for new words enhanced students' processing, thereby making the words more memorable. These findings support Kanellopoulou et al.'s [74] study, which showed that presenting words bimodally can enhance learners' long-term memory.

Moreover, the exercises on the app that generate a higher involvement load ultimately led to a better memory of words. Hulstijn and Laufer [75] indicated that word processing with a higher involvement load is more easily memorized than processing with a lower involvement load. This is consistent with Craik and Lockhart's [76] assertion that the depth of initial processing determines the likelihood of new information being stored in long-term memory.

### 5.3. Students' Perceptions of the App

Results of the questionnaire suggested more than half of the respondents are fond of the app, which was confirmed by 18 out of 19 interviewees expressing their fondness of the app. Moreover, in their journals, most students stated they preferred to learn vocabulary using the app. These results are consistent with Klimova and Polakova [77], who discovered that students had positive attitudes toward vocabulary learning by using apps. The underlying reasons could be attributed to its convenience [78], ease of use and enjoyment [79], usefulness and helpfulness [80], innovation [81], high efficiency [82,83], and satisfaction [56].

## 6. Conclusions

This study contributes to the existing theory of mobile-assisted language learning. Based on the four theories mentioned in Section 2.1, an app was developed especially for CET 4 vocabulary learning among Chinese EFL students. An 8-week experiment was

conducted with two groups. Interviews were analyzed to triangulate the quantitative results. The study revealed that this app successfully enhanced students' vocabulary learning achievements and memory and that students, for the most part, held a favorable opinion of the app.

### 6.1. Pedagogical Implications from the Study

The following three implications are generated from the results of the study. To begin with, educational leaders and policymakers should consider using well-designed mobile apps to promote English learning, such as listening, grammar, and reading. In addition, students should be encouraged to develop their independence when learning vocabulary through apps. Learning autonomy is crucially important in the era of mobile technology, given the limited classroom time for vocabulary learning. As Schmitt noted [84], vocabulary learning and memory require considerable time and effort, as the process is gradual and continuous. Mobile apps can be used to their full advantage to help learners improve their vocabulary independently. It is also noticeable that teachers should supervise students' mobile learning. Moreover, the study's results may provide valuable evidence for educational technology developers and designers, who should consider designing apps that meet each student's needs.

### 6.2. The Limitations of the Research

This study chose its subjects from a limited population located at a particular university in China. Hence, the subjects might not represent their counterparts who study at other universities at different levels, for they are likely to have different English proficiency, campus environments, and demands. Besides, the subjects were sampled from the whole population based on convenience and availability. This sampling method restricts the generalizability of the findings in this study. In addition, the current study is limited in terms of target vocabulary choice which may influence the overall results of the study.

### 6.3. Recommendations for Future Study

The following recommendations for further related research are summarized. First, this study initially attempted to examine vocabulary learning achievement and memory of Chinese EFL learners from a local university through a theory-based app. To further confirm the efficacy of applying an app on EFL learners, it would be necessary to carry out studies on a larger scale, with an increased number of subjects from various grades and universities around the country. Second, an in-depth needs analysis is recommended to identify learners' variations before choosing the target vocabulary. From this study, it was revealed that the two groups held different opinions towards learning the target words per week, which resulted from differences in their vocabulary and English proficiency. Thus, when researchers select target words and design exercises for the design of a future app, they take into consideration the students' language proficiency levels when choosing the target words.

**Supplementary Materials:** The following supporting information can be downloaded at: https://www.mdpi.com/article/10.3390/su15119129/s1, Table S1: Top 80 High-frequency CET4 Words of the Corpus as Target Words; Table S2: Sample of the First Wordlist for the Control Group. Week 2 (10 words); Table S3: Time length, studying time for 10 words/week, and feelings from EG; Section S1: An Excerpt from Vocabulary Size Test (Nation & Beglar, 2007) [43]; Section S2: Knowledge Scale Test of 80 Words; Section S3: Questionnaire of Students' Perceptions of the App; Section S4: The Interview Questions on Perceptions of the App; Section S5: The Informed Consent Form for the Experimental Group; Section S6: The Informed Consent Form for the Control Group; Section S7: Results concerning the Diaries of Experimental Group.

**Author Contributions:** Conceptualization: X.M. and H.L.; software: X.M.; validation: X.M. and H.L.; formal analysis: X.M.; investigation: X.M.; data curation: X.M. and H.L.; writing—original draft preparation: X.M.; writing—review and editing: X.M., H.L. and F.H.; Supervision: H.L. and F.H. All authors have read and agreed to the published version of the manuscript.

**Funding:** This research received no external funding.

**Institutional Review Board Statement:** Not applicable.

**Informed Consent Statement:** Informed consent was obtained from all subjects involved in the study.

**Data Availability Statement:** The authors confirm that the data supporting the findings of this study are available within the article and its appendices.

**Conflicts of Interest:** The authors declare no conflict of interest.

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
