# Peer review of "Exploring the Effects of a Theory-Based Mobile App on Chinese EFL Learners’ Vocabulary Learning Achievement and Memory"

_sustainability, doi:10.3390/su15119129_

Round 1

Reviewer 1 Report

This paper is well-organized and well written. The findings are well discussed and triangulated with relevant methods of data collection. The state of the art is very well described, and the used instruments are also thoroughly validated.

However, the authors can improve the quality of manuscript to a publishable level following the recommendations as below: -

Tile

‘Achievement’ should be added to the title and read as ‘Exploring the Effects of a Theory-Based Mobile App on Chinese EFL Learners’ Vocabulary Learning achievement and Memory’.

Abstract

The demographic information of the two groups should be summarized and stated in the abstract.

The statement made by authors ’As for the learners’ memory, it was discovered that the retention of learned words declined by 10.95% in the experimental group and 27.89% in the control group’ is not appropriated and was not reported in 4. Results

2.1. Theoretical Framework

The authors should describe in detail how the four theories help learners in Vocabulary Learning and Memory so that these explanations can be used to support the writing in 5. Discussion

5. Discussion

The four theories as stated in 2.1. need to be reused to support the discussion of findings.

5.3. Students’ Perceptions of the App

The percentages shouldn’t be restated in this section. Use words such as majority of……..., almost half of the respondents….

6. Conclusions

Implications should be added to the sub-title.  

Author Response

Thanks for your comments. Without your help, the revised manuscript will not be like this. Please see the attachment. 

Reviewer 2 Report

This is an interesting and insightful study, and I think it's publishable in the Journal. Yet, there are some problems to be addressed.

Abstract

1)     A brief description of research design should be included in Abstract.

2)     Better separate Introduction into Introduction and Literature Review

3)     More details about the 4 theories are needed: core ideas, componenets, etc. and how they are related to this research

4)     Table 3-5: why not present t values?

5)     It’s better to compare pre- and post-test performance for the experimental and control groups respectively as well.

6)     It’s better to combine the tables into 1 or 2 tables.

7)     Regarding the questionnaire, isn’t better to analyze the items and group them into different factors and then compute the mean and SD of each factor?

8)     It’s better to discuss the limitations of the research in Conclusion.

9)     More recent publications should be cited.

Author Response

(The authors gave the same response as above.)

Author Response

(The authors gave the same response as above.)

Round 2

Reviewer 3 Report

The revised version is much improved both in theoretic foundations and in  organization of information components.  The reviewer recommends that the manuscript be published after a grammar check.  For example, in last sentence , "So...,thus.." seems not acceptable grammatically. 
